# Use of Levetiracetam in Epileptic Dogs with Chronic Kidney Disease: A Retrospective Study

**DOI:** 10.3390/vetsci8110263

**Published:** 2021-11-03

**Authors:** So-Yeon Gim, Woo-Jin Song, Hwa-Young Youn

**Affiliations:** 1Laboratory of Veterinary Internal Medicine, Department of Veterinary Medical Science, College of Veterinary Medicine, Seoul National University, Seoul 08826, Korea; bewoman@naver.com; 2Laboratory of Veterinary Internal Medicine and Research Institute for Veterinary Science, College of Veterinary Medicine, Jeju National University, Jeju 63243, Korea; 3Research Institute for Veterinary Science, College of Veterinary Medicine, Seoul National University, Seoul 08826, Korea

**Keywords:** adverse effect, chronic kidney disease, dogs, epilepsy, levetiracetam

## Abstract

In human medicine, doses of levetiracetam (LEV) are individualized for patients with epilepsy, depending on the status of the patient’s renal function. However, there are not reports on the individualized dosing of LEV for small animals. The aim of this study is to investigate whether a dose adjustment of LEV is needed in dogs with chronic kidney disease (CKD). Patient databases were searched, and 37 dogs with seizures or epilepsy were retrospectively included in this study. Based on pre-existing CKD, patients were divided into a CKD group (*n* = 20) and a non-CKD group (*n* = 17). We collected kidney panels before and after LEV treatment. Side-effects were monitored for 1 month after the start of LEV administration. In the CKD group, more dogs developed adverse effects (85%) than in the non-CKD group (52.94%). After LEV administration, an increase in blood urea nitrogen and/or serum creatinine was more often reported in the CKD group than it was in the non-CKD group. Our data indicate that in dogs with seizures or epilepsy with pre-existing CKD, an LEV dose-adjustment is needed. During LEV treatment, CKD patients should be monitored for side-effects and may require laboratory evaluation of renal function.

## 1. Introduction

Epilepsy, which affects 50 million people around the world, is the most common neurological disorder [1]. In veterinary medicine, epilepsy is also the most common neurological disorder [2]. The key treatment for this condition is antiepileptic drugs (AEDs), of which the most commonly used are phenobarbital and potassium bromide [3]. However, seizures are not well controlled in 20–30% of dogs with epilepsy [4]. Additionally, dogs can experience severe adverse effects with conventional AED treatment [4]. For these patients, the assessment of new AEDs for the management of epilepsy is essential [5].

Levetiracetam (LEV) is a structurally novel AED that was approved for use in humans as an adjuvant drug for partial-onset seizures in 1999 [6]. LEV is not metabolized via the hepatic cytochrome P450 system [7]. Because of its minimal hepatic metabolism, LEV is favored for use in geriatric or critical patients [7,8,9]. Furthermore, in human medicine, LEV has a minimal effect on the distribution of other AEDs because of its unique metabolic pathway [7,8,9]. Therefore, LEV is particularly favored when initiating polytherapy [10]. Accordingly, LEV has particular uses in special populations [7,8,9]. Based on its encouraging results in treating human epilepsy, levetiracetam is being used frequently in veterinary medicine, especially in dogs [11]. However, there are scant reports of its use in special populations of veterinary patients.

Compared to other AEDs, LEV has a broad margin of safety [7,8]. However, in human medicine, some patients experience undesirable side-effects and toxicity, particularly in special populations, including renal, hepatic, or cardiac insufficiency [12,13]. The usual drug dosage is established for individuals with normal renal function and metabolism [13]. In patients with impaired renal and/or hepatic function, the elimination of drugs may be markedly reduced [9,13]. In such patients, the conventional dosage regimen causes drug accumulation [13,14]. Therefore, in special populations, identifying pharmacokinetic alterations is important to guide prescription strategies [12].

The half-life of LEV elimination is extended to 10–11 h in elderly human patients, and this is also extended in patients with renal disease [12,13]. Therefore, LEV dosing is adjusted in patients with impaired renal function according to the status of the patient’s renal function [12]. Numerous studies in human medicine report individualized LEV dosing according to the status of the patient’s renal function [8,9,12,13]. However, there are not reports concerning individualized LEV dosing according to the status of the patient’s renal status in veterinary medicine. 

Evaluating the use of LEV in preexisting CKD dogs was the goal of this retrospective study, resulting in the need for dose adjustment in patients with kidney disease.

## 2. Materials and Methods

### 2.1. Study Population

For this retrospective study, the patient databases of the Seoul National University Veterinary Medical Teaching Hospital (SNU VMTH) were searched for cases where LEV had been prescribed. The study period was between 5 January 2011 and 28 December 2019. In total, 138 dogs received LEV over a period of 9 years. Hospital records for all eligible patients were reviewed. Information and data necessary for patient evaluation were obtained from the medical records using an electronic chart program (E-friends; pnV, Jeonju, Korea).

### 2.2. Inclusion Criteria

The patient inclusion criteria of this study were as follows: (1) dogs with the presence of seizure or epilepsy and who were treated with oral LEV; (2) who had been diagnosed with or without CKD before LEV administration; (3) and were administered oral LEV for ≥7 days.

### 2.3. Exclusion Criteria

The patient exclusion criteria of this study were as follows: (1) patients who had not been screened for kidney impairments before LEV administration; (2) who were lost to follow up; (3) or who were missing serum blood urea nitrogen (BUN), creatinine, or phosphorus (P) data.

### 2.4. Blood Test Results Analysis

At least 12 h after fasting, blood samples were taken. The components of the blood tests were as follows: BUN, creatinine, P, and symmetric dimethylarginine (SDMA) levels. To determine the effects of LEV on patients’ kidney condition, we compared the blood test results before and after LEV treatment. The LEV-pretreatment blood test results were collected within 1 month prior to starting LEV administration. The LEV post-treatment results were obtained between 2 and 64 days after starting oral LEV administration according to patients’ condition and revisit interval.

Serum BUN was evaluated using standard definitions for clinically significant differences from baseline [15,16]. In patients with normal or low baseline BUN values, a relevant increase in serum BUN was defined as a 25% increase over the upper normal limit, and for patients with initially greater than normal BUN values, a 25% increase higher than baseline was considered [15,16]. Serum creatinine and *p* values were evaluated following the same method as the one used for serum BUN evaluation. 

### 2.5. CKD Evaluation Using the IRIS CKD Diagnosis Guidelines

We investigated whether CKD, defined by the International Renal Interest Society (IRIS) CKD diagnostic guidelines, had been diagnosed before starting LEV administration. CKD staging (CKD stages 1–4) was determined according to the IRIS guidelines for staging CKD [17]. All available information, including serial serum creatinine concentrations, urinalysis, and diagnostic imaging findings, were used for the diagnosis of CKD and for the determination of the CKD stage, before starting LEV administration [17]. 

### 2.6. Epilepsy Frequency and Days Prior to and during LEV Treatment

The data collected from the patient records included age and weight at the start of LEV treatment, the total number and days of epilepsy prior to starting LEV administration, and the total number and days of epilepsy occurring during LEV treatment. The total number of epilepsy incidents during LEV treatment that occurred within 1 month after starting LEV administration was counted [18]. Additionally, the total number of days on which epilepsy occurred during the 1 month after starting LEV treatment was counted [18].

### 2.7. Levetiracetam Administration

All of the patients in this study were given a daily maintenance dose of LEV (Keppra; UCB Pharma SA, Belgium; Levetiracetam; Rhino Bio, Daejeon, Korea) per os (PO), depending on the patient’s status, and side-effects and therapeutic responses were monitored. LEV was administered at an initial dose of 10.0–45.0 mg/kg PO, q8–12h. The LEV dose was adjusted based on clinical response. LEV was used as the primary or as an add-on therapy, depending on the patient’s conditions. When used as add-on therapy, LEV was used in combination with phenobarbital, potassium bromide (KBr), zonisamide, or diazepam.

Side-effects of oral LEV were recorded; in particular, whether the following variable symptoms were more frequently provoked during LEV treatment was monitored: ataxia, sedation, decreased appetite, polydipsia, vomiting, diarrhea, hypersalivation, and behavior changes (showing aggression) [10,11,18]. The side-effects that occurred during LEV treatment were monitored for 1 month after the start of LEV.

### 2.8. Statistical Analysis

Statistical analysis was performed using commercial software (R package, ver. 3.1.1; The R Foundation for Statistical Computing, Vienna, Austria). Analytical data are presented as mean ± standard deviation (SD). Data are presented as median with range and SD. Differences between the variables from the CKD group and the non-CKD group were tested using Fisher’s exact test or the chi-square test. *p*-values < 0.05 were regarded significant. 

## 3. Results

### 3.1. Study Animals 

In total, 138 dogs with seizure or epilepsy were given oral LEV over a period of 9 years, of which 62 patients met the inclusion criteria. A further 25 patients were excluded based on the exclusion criteria. Finally, 37 dogs were included in this study. We divided these patients into a CKD group (*n* = 20) and a non-CKD group (*n* = 17).

The CKD group included 20 dogs with a mean age of 12.53 ± 3.78 years (range, 2.58–17.5 years) and a median weight of 5.03 ± 2.78 kg (range, 1.8–13.1 kg) when treatment with LEV began (Table 1). Several breeds were included, with the Maltese being the most common (20%) (Table 1). Dogs included both intact and neutered females and males (Table 1). The CKD group’s etiological diagnoses are presented in Table 2. 

The non-CKD group consisted of 17 dogs with a mean age of 10.26 ± 3.73 years (range, 2.08–17.25 years) and a median weight of 5.44 ± 5.56 kg (range, 1.25–25.5 kg) when LEV treatment began (Table 1). Several breeds were recorded, with Maltese once again being the most frequent (Table 1), and both intact and neutered males and females were included (Table 1). The non-CKD group’s etiological diagnoses are also presented in Table 2. 

### 3.2. Pre-Existing CKD Evaluation Based on IRIS CKD Diagnostic Guidelines

In the CKD group, most patients (60%) were CKD stage 1, while the remainder were CKD stage 2; there were no CKD stage 3 or 4 patients (Table 1).

### 3.3. LEV-Based Therapy in Both Groups

All of the patients in this study were given LEV dosages of 10.0–45.0 mg/kg orally, q8–12h. Most patients (*n* = 33, 89.19%) did not receive a loading dose. The mean duration of the oral LEV therapy in this study was 263.62 ± 547.81 days (range, 2–2649 days days). 

In the CKD group, the initial LEV dose was 21.94 ± 8.36 mg/kg (range, 10.0–45.0 mg/kg) PO, q8–12h. In the non-CKD group, this dose was 20.58 ± 8.72 mg/kg (range 10.0–30.0 mg/kg) PO, q8–12h (Table 2). The mean duration of the oral LEV therapy in the CKD group was 147.85 ± 183.54 days (range, 8–613 days), and in the non-CKD group it was 399.82 ± 761.05 days (range, 2–2649 days) (Table 2). 

Of the 20 CKD patients, 10 patients (50%) received LVE monotherapy, while the others received LEV in combination with phenobarbital, KBr, zonisamide, and diazepam. Four patients (20%) received LEV in combination with phenobarbital, two patients (10%) received LEV in combination with zonisamide, one patient (5%) received LEV in combination with diazepam, one patient (5%) received LEV in combination with phenobarbital and zonisamide, one patient (5%) received LEV in combination with phenobarbital and potassium bromide, and one patient (5%) used all 5 drugs simultaneously. 

Of the 17 non-CKD patients, 9 patients (52.94%) received LEV monotherapy. Two patients (11.76%) received LEV in combination with phenobarbital, one patient (5.88%) received LEV in combination with zonisamide, one patient (5.88%) received LEV in combination with phenobarbital and zonisamide, one patient (5.88%) received LEV in combination with phenobarbital and diazepam, two patients (11.76%) received LEV in combination with zonisamide and gabapentin, and one patient (5.88%) received LEV in combination with phenobarbital, KBr, and zonisamide simultaneously. No drug interactions with LEV were identified among the study patients. 

### 3.4. Outcomes of Patients on LEV Therapy

In the CKD group, the mean number of seizures prior to starting LEV was 59.7 ± 161.7 (range, 0–735), and the mean number of seizure days before LEV was 17.55 ± 30.96 (range, 0–122) (Table 3). In the non-CKD group, the mean number of seizures prior to starting LEV was 42.59 ± 71.313 (range, 1–275), and the mean number of seizure days prior to LEV treatment was 16.65 ± 25.72 (range, 1–89) (Table 3). The total number of seizures and the total days of seizures during LEV treatment during the 1 month after the start of LEV treatment was 13.55 ± 38.89 (range, 0–162) and 2.8 ± 7.05 (range, 0–30), respectively, in the CKD group (Table 3). In the non-CKD group, during LEV treatment, the total mean number of seizures while on LEV treatment was 7.18 ± 12.60 (range, 0–48), and the total mean number of seizure days while on LEV treatment was 1.59 ± 1.94 (range, 0–6) (Table 3).

In general, the side-effects of LEV include ataxia, sedation, anorexia, polydipsia, vomiting, diarrhea, hypersalivation, and behavior changes (showing aggression). Of the 37 patients, 26 patients (70.27%) showed adverse effects during LEV administration. Most patients showed more than one side-effect at the same time (Table 4). More dogs in the CKD group were reported to have adverse effects than in the non-CKD group, and there was a statistically difference between the two groups (85% vs. 52.94%, *p* = 0.033). In the CKD group, a significant increase in sedation was identified during LEV treatment compared to baseline, but it was not statistically significant compared to the non-CKD group (55 versus 41.18%, *p* = 0.40). No dogs showed life-threatening adverse effects.

### 3.5. Clinically Relevant Increases in Serum BUN, Creatinine, and Phosphorus after LEV Administration

When all of the eligible patients were considered, the occurrence of “clinically relevant increases” in serum BUN, which consider the magnitude of the increase in relation to the baseline value, was statistically significantly different between the CKD group and the non-CKD group (*p* = 0.028; Table 5). The incidence of clinically relevant increases in serum creatinine (*p* = 0.003; Table 5) and in both serum BUN and creatinine concurrently (*p* = 0.014, Table 5) was statistically significantly different between the groups. However, there was no significant difference in the occurrence of clinically relevant increases in serum phosphorus between the groups (*p* = 0.50; Table 5).

## 4. Discussion

In human patients with impaired renal function, levetiracetam (LEV) dosing is individualized depending on the status of the patient’s renal function [7,9,12,13], but in veterinary medicine, there have been no reports on individualized LEV dosing according to the status of the patient’s renal function to date, and there have been few studies on the use of LEV in particular dog populations [19,20]. Thus, the impact of oral LEV on dogs with pre-existing CKD has not been previously reported; the present study provides insights into this matter.

The initial LEV dose was 21.94 ± 8.36 mg/kg (range, 10.0–45.0 mg/kg) PO, q8–12h in the CKD group, and 20.58 ± 8.72 mg/kg (range 10.0–30.0 mg/kg) PO, q8–12h, in the non-CKD group in this study. In humans, the elimination half-life of LEV is extended in patients with renal impairment [7,8,9,12,13], and LEV dosing is therefore adjusted according to the status of the patient’s renal function [8,9,12]. Pharmacokinetic studies suggest a dose of LEV 20 mg/kg, q8h to obtain plasma concentrations in normal dogs, which is similar to clinically significant plasma concentrations in humans [21,22]. Consequently, these studies suggest that dogs with renal impairments should be given a LEV dose < 20 mg/kg, q8h [21,22]. It is estimated that some of our CKD patients had higher LEV concentrations than the therapeutic range, as they were administered LEV at doses that were more than 20 mg/kg, q8h [21,22]. We did not obtain serum concentrations of LEV. However, based on previous studies, we assumed that LEV serum concentrations might be higher in the CKD group than in the non-CKD group, which was also concordant with our other study results. Statistically significantly higher number of dogs in the CKD group were reported to have side-effects than those in the non-CKD group (85% vs. 52.94%, *p* = 0.033). Further study is necessary to determine individualized LEV dosing according to the status of a dog’s renal function.

In comparison to other AEDs, LEV has a broad margin of safety [7,8]. In this study, LEV was used as add-on therapy or primary therapy. Many patients in the current study had concurrent diseases because they were middle-aged or older. Because of the minimal hepatic metabolism, LEV is favored for geriatric or critical human patients [7,8,9]. Furthermore, in human medicine, LEV has been shown to have minimal effects on the distribution of other AEDs due to its unique metabolic pathway [7,8,9]. Therefore, the use of LEV in specific complex medical situations, particularly in polytherapy, has been encouraged [10]. In this study, we found that LEV was mostly well tolerated in dogs, but it was not free from side-effects.

In general, the side-effects of LEV include ataxia, sedation, anorexia, polydipsia, vomiting, diarrhea, hypersalivation, and behavior changes (aggression) [10,11,18]. In this study, most patients showed more than one side-effect simultaneously. In the CKD group, there was a significant increase in sedation during LEV administration compared to at baseline, but this increase was not remarkably different from the one observed in the non-CKD group (*p* = 0.40). Because of sedation, chronic recurrent dehydration associated with periodic water intake may occur [23], which can be problematic, particularly in CKD patients. A key management factor in CKD patients is access to fresh water anytime because the ability to concentrate urine and excrete water is impaired in CKD patients [24]. If a patient with CKD is sedated, water intake may be irregular, which may accelerate dehydration [23], decreasing blood flow to the kidneys [24]. Consequently, uremic toxins are retained [24]. This was supported by the results of the present study. In the CKD group, nine patients (45%) demonstrated clinically significant increases [15,16] in serum BUN (Table 5) and eight demonstrated clinically relevant increases in serum creatinine (Table 5). Moreover, in that group, six patients (30%) had concurrent clinically significant increases [15,16] in serum BUN and creatinine (Table 5). On the other hand, in the non-CKD group, there were no clinically significant increases in BUN and/or creatinine (Table 5). Therefore, canine CKD patients who are administered LEV should be monitored, and they may require physical and/or laboratory evaluation for dehydration [25,26].

The statically significant difference in the incidence of clinically relevant increases in serum BUN and/or creatinine between the CKD and non-CKD groups (Table 5) may be due to the acute kidney injury (AKI) that is superimposed on pre-existing CKD. In human medicine, although it has been shown that kidney function can fully recover after overcoming an AKI, cumulative observational data have revealed that AKI can worsen underlying CKD [27]. There have been four case reports in humans where LEV was identified as a possible contributor to AKI [25,26,28,29]. Based on those reports, patients taking LEV who present with abdominal pain, oliguria, nausea/vomiting, and a high creatinine concentration may require laboratory assessment for hidden renal adverse effects [25,26], and renal function during treatment with LEV should be monitored [25,26,28,29]. In the differential diagnosis for any unexplained acute renal failure, AKI due to LEV should be considered [28]. However, in veterinary medicine, there have been no reports of renal toxicity attributed to LEV, and further studies are necessary to assess renal toxicity that is secondary to LEV in dogs.

As in other retrospective studies, this study had some limitations. First, we had some missing values, such as serum SDMA values, which increase during CKD progression and that correlate strongly with a decrease in the glomerular filtration rate [30,31], because the data were collected retrospectively and SDMA has only been included in the diagnostic criteria for the diagnosis of CKD since 2015 [32]. We rarely measured SDMA values before this time-point. Therefore, future studies including SDMA results are needed. Second, in this study, the patients were chronically ill because they were middle-aged or older. Consequently, it is possible that the CKD stage might have been falsely high or falsely low [17,24]; the CKD stage of the patients with prerenal azotemia could have been falsely high [17,24], and that of patients who had lost muscle mass could have been falsely low [17,24]. In retrospective studies, it is difficult to solve this issue. Third, we did not measure the plasma LEV concentration values, as there is no therapeutically effective LEV concentration known in dogs [10]. Therefore, in practice, LEV concentration monitoring is rarely performed [10]. However, based on previous studies, it could be assumed that the CKD group had higher LEV concentrations than the non-CKD group did [7,8,9,21,22]. Lastly, our data could not exclude the possibility that other AEDs (such as phenobarbital, potassium bromide, zonisamide, or diazepam) contributed to anti-epileptic effects or adverse effects. It was also difficult to control the dosage of LEV, owner compliance, and factors in the living environment, such as diet. Further prospective and controlled studies are needed.

## 5. Conclusions

In conclusion, no previous study has evaluated the use of oral LEV in canine patients with pre-existing CKD. We showed that dogs in the CKD group had more side-effects than those in the non-CKD group. In the CKD group, the incidence of clinically significant increases in serum BUN and/or creatinine was more frequent than in the non-CKD group. Although this study had some limitations, as is sometimes the case in retrospective studies, the findings presented here may assist in the treatment of many epileptic dogs with CKD, resulting in the proper dosing of LEV in animals with pre-existing renal impairment.

## Figures and Tables

**Table 1 vetsci-08-00263-t001:** Characteristics of patients with the CKD group/non-CKD group in the present study.

	CKD Group (*n* = 20)	Non-CKD Group (*n* = 17)
Variables	No. of Dogs (%)	No. of Dogs (%)
CKD stage before LEV administration		
non-CKD	0 (0%)	17 (100%)
CKD stage 1	12 (60%)	0 (0%)
CKD stage 2	8 (40%)	0 (0%)
CKD stage 3 or 4	0 (0%)	0 (0%)
Breeds		
Maltese	4 (20%)	7 (41.18%)
Yorkshire Terrier	2 (10%)	2 (11.76%)
Pekingese	2 (10%)	0 (0%)
Cocker Spaniel	2 (10%)	2 (11.76%)
Poodle	2 (10%)	0 (0%)
Mixed breed	2 (10%)	3 (17.65%)
Chihuahua	0 (0%)	2 (11.76%)
Afghan Hound	0 (0%)	1 (5.88%)
Sex		
Male	3 (15%)	4 (23.53%)
Male castrated	3 (15%)	4 (23.53%)
Female	4 (20%)	1 (5.88%)
Female spayed	10 (50%)	8 (47.06%)
Body weight at start of LEV, kg		
Mean ± SD (range)	5.03 ± 2.78 kg (range, 1.8–13.1 kg)	5.44 ± 5.56 kg (range, 1.25–25.5 kg)
Age at start of LEV, years (min–max years)		
Mean ± SD (range)	12.53 ± 3.78 years (range, 2.58–17.5 years)	10.26 ± 3.73 years (range, 2.08–17.25 years)

**Table 2 vetsci-08-00263-t002:** LEV-based therapy in dogs of the CKD group/non-CKD group.

	CKD Group (*n* = 20)	Non–CKD Group (*n* = 17)
Variables	No. of Dogs (%)	No. of Dogs (%)
Indications of LEV		
Undiagnosed epilepsy	11 (55%)	6 (35.30%)
Idiopathic epilepsy	3 (15%)	3 (17.65%)
Hydrocephalus	1 (5%)	3 (17.65%)
Hydrocephalus with syringohydromyelia	2 (10%)	1 (5.88%)
Chiari-like malformation	0 (0%)	2 (11.76%)
Meningoencephalitis of unknown etiology	1 (5%)	0 (0%)
Hydrocephalus with meningoencephalitis of unknown etiology	0 (0%)	1 (5.88%)
Geriatric vestibular disease suspected	1 (5%)	0 (0%)
Reactive epilepsy	1 (5%)	0 (0%)
Brain tumor	0 (0%)	1 (5.88%)
Variables	Value	Value
Initial dose of LEV administered		
Mean ± SD (range)	21.94 ± 8.36 mg/kg (range, 10.0–45.0 mg/kg)	20.58 ± 8.72 mg/kg (range 10.0–30.0 mg/kg)
Duration of oral LEV therapy (days)		
Mean ± SD (range)	147.85 ± 183.54 days (range 8–613 days)	399.82 ± 761.05 days (range 2–2649 days)

**Table 3 vetsci-08-00263-t003:** Seizure frequency of the CKD group/non–CKD group.

	CKD Group (*n* = 20)	Non–CKD Group (*n* = 17)
Variables	Value	Value
Epilepsy frequency prior to LEV treatment		
Mean ± SD (range)	59.7 ± 161.70 (range, 1–735)	42.59 ± 71.31 (range, 1–275)
Days of epilepsy prior to LEV treatment		
Mean ± SD (range)	17.55 ± 30.96 (range, 1–122)	16.65 ± 25.72 (range, 1–89)
Epilepsy frequency during LEV treatment		
Mean ± SD (range)	13.55 ± 38.88 (range, 0–162)	7.18 ± 12.60 (range, 0–48)
Days of epilepsy during LEV treatment.		
Mean ± SD (range)	2.8 ± 7.05 (range, 0–28)	1.59 ± 1.94 (range, 0–6)

**Table 4 vetsci-08-00263-t004:** Side-effects of the oral LEV were observed in the CKD group/non–CKD group.

	CKD Group (*n* = 20)	Non–CKD Group (*n* = 17)	
Variables	No. of Dogs (%)	No. of Dogs (%)	*p*-Value
Any adverse event	17 (85%)	9 (52.94%)	0.033
Sedation	11 (55%)	7 (41.18%)	0.40
Ataxia	5 (25%)	3 (17.65%)	NA
Anorexia	9 (45%)	2 (11.76%)	NA
Polydipsia	2 (10%)	0 (0%)	NA
Vomiting	7 (35%)	1 (5.88%)	NA
Diarrhea	3 (15%)	2 (11.76%)	NA
Hypersalivation	3 (15%)	0 (0%)	NA
Behavior changes (showing aggression)	1 (5%)	0 (0%)	NA

**Table 5 vetsci-08-00263-t005:** Clinically relevant serum BUN, creatinine, and phosphorus increase in the CKD group/non–CKD group.

	CKD Group (*n* = 20)	Non–CKD Group (*n* = 17)	
Variables	No. of Dogs (%)	No. of Dogs (%)	*p*-Value
BUN elevation	9 (45%)	2 (11.76%)	0.028
Phosphorus elevation	2 (10%)	0 (0%)	0.50
Creatinine elevation	8 (40%)	0 (0%)	0.003
BUN and creatinine elevation	6 (30%)	0 (0%)	0.014

## Data Availability

The data presented in this study are contained within the article.

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
