# Peer review of "Use of Levetiracetam in Epileptic Dogs with Chronic Kidney Disease: A Retrospective Study"

_vetsci, 2021, doi:10.3390/vetsci8110263_

Round 1

Reviewer 1 Report

In the current review manuscript “vet sci- 1400052”, the authors conduct a retrospective study from the patient databases to understand if the dosing of levetiracetam (LEV) for small animals was warranted in connection to Chronic Kidney Disease (CKD). The topic is generally of interest to the readers of the journal yet the results don’t appear to be adequately sound.

There are several fundamental deficiencies in the manuscript and the results could be potentially influenced by high variability within the sample groups and measures. Therefore, there are several factors that would result in erroneous interpretation. The authors describe some of these limitations. Nevertheless, additional limitations such as dosing variabilities, concurrent dosing with other AEDs, the incidence of other comorbidities, diets, owner compliance are yet to be discussed.  

Validation of these results in a replication cohort in a larger cohort is highly desirable and will improve the overall soundness of the manuscript and that the findings are legitimate.  

Author Response

Reveiwer #1

1. COMMENT: In the current review manuscript “vet sci- 1400052”, the authors conduct a retrospective study from the patient databases to understand if the dosing of levetiracetam (LEV) for small animals was warranted in connection to Chronic Kidney Disease (CKD). The topic is generally of interest to the readers of the journal yet the results don’t appear to be adequately sound.

RESPONSE: Thank you for your valuable comments. Herein, we explain how we revised the paper based on those comments and recommendations. The manuscript has certainly benefited from these revision suggestions.

2. COMMENT: There are several fundamental deficiencies in the manuscript and the results could be potentially influenced by high variability within the sample groups and measures. Therefore, there are several factors that would result in erroneous interpretation. The authors describe some of these limitations. Nevertheless, additional limitations such as dosing variabilities, concurrent dosing with other AEDs, the incidence of other comorbidities, diets, owner compliance are yet to be discussed.

RESPONSE: Thank you for raising this important point. We could not measure plasma LEV concentration values, as there is no therapeutically effective LEV concentration known in dogs. However, based on previous studies, it could be assumed that the CKD group had higher LEV concentrations than the non-CKD group. In addition, our data could not exclude the possibility that other AEDs contributed to adverse effects. It was also difficult to control the dosage of LEV, owner compliance, and living environment such as diet. It has been described in the revision. We hope our approach acceptable.

3. COMMENT: Validation of these results in a replication cohort in a larger cohort is highly desirable and will improve the overall soundness of the manuscript and that the findings are legitimate.

RESPONSE: Thank you for your raising this important point. We agree that further prospective and controlled studies are needed. It has been described in the revision. We hope our approach acceptable.

Reviewer 2 Report

Reviewer comments for manuscript ID vet sci-1400052 entitled ‘Use of Levetiracetam in Epileptic Dogs with Chronic Kidney Disease: a Retrospective Study’

General comments

It is an interesting study in the slow evolving therapeutics of canine epilepsy. Pentobarbital and Potassium bromide have been the mainstay of canine epilepsy therapy and newer drugs are direly needed for better and faster treatment of the disease. It is a good attempt in analysing the treatment regimes in retrospect and the limitations of such studies have been acknowledged by the authors. I have concerns on the grouping of the patients as patients on multi-therapy and levetiracetam alone have been grouped together. It is obvious patients on multi-therapy will show higher serum BUN and creatinine values than patients on single drug therapy, then I wonder how the effect of tis drug was evaluated in multi-therapy patients.

Discussion needs more in-depth analysis and clearly indicate what future studies could be dovetailed from this work.

Specific comments

Introduction

Lines 42-45: Please reframe these lines for more clarity to the reader. I suggest ‘Based on its encouraging results in treating human epilepsy, levetiracetam is being frequently used in veterinary medicine, especially in dogs [11]. However, there are scant reports of its use in special populations of veterinary patients.

Line 47: I think it will be better to clarify ‘special populations’ for the readers here at the outset.

Lines 84-86: Please justify/clarify the time intervals for the pre-and post-treatment blood sampling for serum biochemistry.

Lines 113-114: Using LEV with other drugs such as diazepam, potassium bromide and pentobarbital should alter the creatinine and BUN levels in a different way than giving it alone. How do you justify grouping these types of patients in one group? Furthermore, there are reports of neutering having effects on epilepsy. Please clarify.

Table 2: Please clarify, how did you differentiate between undiagnosed seizure and undiagnosed epilepsy.

Lines 204-06: Please delete.

Line 2725-77: Please discuss the significance of the measurement of serum SDMA here.

Author Response

Reveiwer #2

General comments

1. COMMENT: It is an interesting study in the slow evolving therapeutics of canine epilepsy. Pentobarbital and Potassium bromide have been the mainstay of canine epilepsy therapy and newer drugs are direly needed for better and faster treatment of the disease. It is a good attempt in analysing the treatment regimes in retrospect and the limitations of such studies have been acknowledged by the authors. I have concerns on the grouping of the patients as patients on multi-therapy and levetiracetam alone have been grouped together. It is obvious patients on multi-therapy will show higher serum BUN and creatinine values than patients on single drug therapy, then I wonder how the effect of tis drug was evaluated in multi-therapy patients. Discussion needs more in-depth analysis and clearly indicate what future studies could be dovetailed from this work.

RESPONSE: Thank you for your valuable comments. Herein, we explain how we revised the paper based on those comments and recommendations. The manuscript has certainly benefited from these revision suggestions.

Specific comments

2. COMMENT: Lines 42-45: Please reframe these lines for more clarity to the reader. I suggest ‘Based on its encouraging results in treating human epilepsy, levetiracetam is being frequently used in veterinary medicine, especially in dogs [11]. However, there are scant reports of its use in special populations of veterinary patients.

RESPONSE: Thank you for your suggestion. We revised the sentence as your comment in the revision.

3. COMMENT: Line 47: I think it will be better to clarify ‘special populations’ for the readers here at the outset.

RESPONSE: Thank you for your detailed comment. We revised the sentence as follow; some patients experience undesirable side-effects and toxicity, particularly in special populations including renal, hepatic, or cardiac insufficiency. We hope our approach acceptable.

4. COMMENT: Lines 84-86: Please justify/clarify the time intervals for the pre-and post-treatment blood sampling for serum biochemistry.

RESPONSE: Thank you for raising this important point. The LEV post-treatment results were obtained between 2 and 64 days after starting oral LEV administration according to patients’ condition and revisit interval. It has been described in the revision. We hope our approach acceptable.

5. COMMENT: Lines 113-114: Using LEV with other drugs such as diazepam, potassium bromide and pentobarbital should alter the creatinine and BUN levels in a different way than giving it alone. How do you justify grouping these types of patients in one group? Furthermore, there are reports of neutering having effects on epilepsy. Please clarify.

RESPONSE: Thank you for raising this important point. As retrospective study, it is one of limitations that our data could not exclude the possibility that other AEDs contributed to adverse effects. It was also difficult to control the dosage of LEV, owner compliance, and living environment such as diet. Also, we could not measure plasma LEV concentration values, as there is no therapeutically effective LEV concentration known in dogs. However, based on previous studies, it could be assumed that the CKD group had higher LEV concentrations than the non-CKD group. It has been described in the revision. We hope our approach acceptable.

6. COMMENT: Table 2: Please clarify, how did you differentiate between undiagnosed seizure and undiagnosed epilepsy.

RESPONSE: Thank you for raising this important point. We deleted ‘undiagnosed seizure’ and added to ‘undiagnosed epilepsy’, because we could not differentiate them. We hope our approach acceptable.

7. COMMENT: Lines 204-06: Please delete.

RESPONSE: Thank you for your detailed comment. As your comment, we deleted the sentence in the revision.

8. COMMENT: Line 2725-77: Please discuss the significance of the measurement of serum SDMA here.

RESPONSE: Thank you for your detailed comment. Serum SDMA increase during CKD progression correlating strongly with a decrease in glomerular filtration rate (Nabity et al., Symmetric Dimethylarginine Assay Validation, Stability, and Evaluation as a Marker for the Early Detection of Chronic Kidney Disease in Dogs. Journal of Veterinary Internal Medicine 2015; Pelander et al., Comparison of the diagnostic value of symmetric dimethylarginine, cystatin C, and creatinine for detection of decreased glomerular filtration rate in dogs. Journal of Veterinary Internal Medicine 2019). It has been described in the revision.

Reviewer 3 Report

Dear authors,

This study analyses different doses of levetiracetam in epileptic dogs with chronic kidney disease, because this treatment in humans is adjusted according to the state of the patient's kidneys, but in the veterinary clinic it is not done yet. The results ¡ are interesting and very useful for veterinary clinical practice. The manuscript is well written and structured, the introduction provides sufficient background, the research design is appropriate, the methods are adequately described, and the results are clearly presented. However, from my point of view some changes could substantially improve the manuscript.

  • Abbreviations that appear in the text must be explained the first time they appear in the text (for example, lines 16, 21, etc).
  • Lines 46-53: authors should explain whether this paragraph refers to humans or dogs.
  • In discussion, it is missed that the authors analyze the possible effects of multidrug treatment, since some of their patients receive other drugs in addition to LEV. It is also missed that they compare the adverse effects found with adverse effects that are already known from treatment with LEV.

Author Response

Reviewer #3

1. COMMENT: This study analyses different doses of levetiracetam in epileptic dogs with chronic kidney disease, because this treatment in humans is adjusted according to the state of the patient's kidneys, but in the veterinary clinic it is not done yet. The results ¡ are interesting and very useful for veterinary clinical practice. The manuscript is well written and structured, the introduction provides sufficient background, the research design is appropriate, the methods are adequately described, and the results are clearly presented. However, from my point of view some changes could substantially improve the manuscript.

RESPONSE: Thank you for your valuable comments. Herein, we explain how we revised the paper based on those comments and recommendations. The manuscript has certainly benefited from these revision suggestions.

2. COMMENT: Abbreviations that appear in the text must be explained the first time they appear in the text (for example, lines 16, 21, etc).

RESPONSE: Thank you for your detailed comment. We explained the abbreviations in the abstract section as your comment in the revision.

3. COMMENT: Lines 46-53: authors should explain whether this paragraph refers to humans or dogs.

RESPONSE: Thank you for your detailed comment. This paragraph refers to humans, and it has been described in the revision.

4. COMMENT: In discussion, it is missed that the authors analyze the possible effects of multidrug treatment, since some of their patients receive other drugs in addition to LEV. It is also missed that they compare the adverse effects found with adverse effects that are already known from treatment with LEV.

RESPONSE: Thank you for raising this important point. As retrospective study, it is one of limitations that our data could not exclude the possibility that other AEDs contributed to adverse effects. It was also difficult to control the dosage of LEV, owner compliance, and living environment such as diet. Also, we could not measure plasma LEV concentration values, as there is no therapeutically effective LEV concentration known in dogs. However, based on previous studies, it could be assumed that the CKD group had higher LEV concentrations than the non-CKD group. It has been described in the revision. We hope our approach acceptable.

Round 2

Reviewer 1 Report

The authors have attempted to adress the comments raised. The authors also acknowledged more limiations of the study.  Although the authors did not conduct some of the experiments/analysis indicated, reasonable justification was provided with an intention that  future studies will need to address this issue. However, it is important to indicate this statement in the conclusion also. After addressing this, I recommend this manuscript may be considered for publication. 

Author Response

Thank you for your valuable comments. And, this manuscript has certainly benefited from these revision suggestions. We revised the conclusion section in the revision as your comment; ‘Although this study had some limitations as in other retrospective studies, the findings presented here may assist in the treatment of many epileptic dogs with CKD, resulting in proper dosing of LEV in animals with pre-existing renal impairment.’

Reviewer 2 Report

Reviewer comments on manuscript ID manuscript ID vet sci-1400052 entitled ‘Use of Levetiracetam in Epileptic Dogs with Chronic Kidney Disease: a Retrospective Study- Round -2

General comments

I congratulate the authors for revising the document extensively. There are some serious flaws in the research design that must be highlighted in the discussion and conclusions of the study as it is a retrospective study. For example refer to my comment Lines 113-14 'Using LEV with other drugs such as diazepam, potassium bromide and pentobarbital should alter the creatinine and BUN levels in a different way than giving it alone. How do you justify grouping these types of patients in one group? Furthermore, there are reports of neutering having effects on epilepsy.

Please emphasize that these are the limitations of the study and please quote references  to justify the effects of other adjunct therapies and neutering on epilepsy. 

Author Response

Thank you for your valuable comments. And, this manuscript has certainly benefited from these revision suggestions.

First, we revised the conclusion section in the revision as your comment; ‘Although this study had some limitations as in other retrospective studies, the findings presented here may assist in the treatment of many epileptic dogs with CKD, resulting in proper dosing of LEV in animals with pre-existing renal impairment.

Second, we revised the discussion section in the revision as your comment; ‘Lastly, our data could not exclude the possibility that other AEDs (such as phenobarbital, potassium bromide, zonisamide, or diazepam) contributed to anti-epileptic effects or ad-verse effects. It was also difficult to control the dosage of LEV, owner compliance, and liv-ing environment such as diet. Further prospective and controlled studies are needed.’

We hope our approach acceptable.

Reviewer 3 Report

Dear authors,

The manuscript has improved and, from my point of view, it can be published in its current form.

Author Response

Thank you for your kind comments. The manuscript has certainly benefited from your revision suggestions.